# A Comparison Study between Universal Design for Learning-Multiple Intelligence (UDL-MI) Oriented STEM Program and Traditional STEM Program for Inclusive Education

**Nurfarahin Nasri, Nik Mohd Rahimi, Nurfaradilla Mohamad Nasri *** and **Mohamad Asyraf Abd Talib**

Faculty of Education, Universiti Kebangsaan Malaysia, Bangi 43600, Selangor, Malaysia;
nurfarahinnasri@yahoo.com (N.N.); nik@ukm.edu.my (N.M.R.); asyraf_717@yahoo.com (M.A.A.T.)
**\*** Correspondence: nurfaradilla@ukm.edu.my

**Abstract:** The Universal Design for Learning (UDL) model and Multiple Intelligence (MI) theory hold considerable promise for advancing inclusive education. However, the effectiveness of UDL and MI in supporting inclusive science, technology, engineering, and mathematics (STEM) education has never been tested empirically. This study examined the impact of the UDL-MI-oriented STEM program on eighth-grade rural students' attitudes towards STEM through mixed research design. A total of 122 students were selected through purposive sampling and randomly distributed into experimental (N = 62) and control groups (N = 60). The experimental group experienced STEM learning through the UDL-MI-oriented program and the control group received traditional classroom teaching. Both groups studied learning units on environmental sustainability and conservation for 10 weeks. Changes in students' attitudes towards STEM were assessed over the time period (pre-test, post-test, follow-up) using an adapted Mahoney Student Attitude toward STEM instrument and individual interviews (post-test, follow-up). The results indicated that in comparison to the traditional classroom, the UDL-MI-oriented program significantly improved and maintained the students' attitudes towards STEM. Next, the qualitative findings were presented to support the statistical analyses. This study provides empirical evidence for adopting the UDL-MI-oriented STEM program as an effective way of providing inclusive STEM education to rural students by enhancing their attitudes towards STEM.

**Keywords:** inclusive education; Universal Design for Learning (UDL); Multiple Intelligence (MI); inclusive science; technology; engineering and mathematics (STEM) education



## 1. Introduction

Ensuring that every citizen has an equal opportunity to achieve educational success remains a global challenge. The Sustainable Development Goal 4 on Quality Education and the Education 2030 Framework for Action have played important roles in promoting inclusive, and equitable education for the development of a more cohesive society. In this context, inclusive education is defined as both approach and process of respecting the diversity of needs, abilities, and characteristics of all students by removing any barrier that could potentially limit their academic achievement [1]. Among the underserved, marginalized, and vulnerable groups of students, UNESCO pays special attention to rural students as they comprise the majority of those who experience exclusion within and from mainstream education [2].

Over the past three decades, research has shown that students who experience inclusive science, technology, engineering, and mathematics (STEM) education attain higher academic achievement and improvement of scientific skills. However, the inclusion of rural students in STEM education is seriously impeded by various pedagogical challenges [3]. This is especially true as many researchers have documented STEM teachers' limited knowledge and experience to cater to the learning needs of rural students [4–6]. Accordingly, several

models and frameworks highlighting the core principles of inclusive STEM education have been proposed, such as (a) Rosser's [7] female-friendly model, (b) Banks' [8] multiculturalism instructional model, (c) Kirch et al.'s [9] CLASS model, and (d) Kolonich, Richmond, and Krajcik's [10] inclusive instructional framework. Critical analysis of these models and frameworks reveals a common key feature of inclusive STEM education—the employment of multiple instructional approaches and assessment strategies that could meet the diverse students' learning needs and capabilities.

Despite much advancement in the field of inclusive STEM education, STEM enrolment rates continue to decline across many countries [11]. The findings from PISA-2015 have also found that the students' attitude towards STEM is relatively low, where only 40% of the student population are interested to have STEM-related careers. Students' attitude which refers to a general feeling towards a certain issue, learning process, and person, is fundamental to promote the students' desire to pursue their future learning in STEM [12]. In this instance, many STEM educators have therefore focused on finding the most appropriate type of teaching method that can help to shape positive students' attitudes towards STEM [12]. Inclusive pedagogical strategies emerge as one of the potential alternatives to enhance student's attitudes towards STEM by specifically addressing their cognitive, emotional, and social needs [13].

Amid the global movement of achieving accessible and quality education for all, more systemic and intensive efforts are organized to make STEM learning more inclusive for all students [14]. From this context, this study adapted both the Universal Design for Learning (UDL) model and Multiple Intelligence (MI) theory to design an inclusive STEM program, specifically for rural students. UDL which serves as a proactive approach to designing an inclusive learning environment, acknowledges student's diversity and prompts the teachers to responsively meet various students' learning needs [15]. UDL encompasses three core principles: (a) multiple means of engaging students, (b) multiple means of representing knowledge, and (c) multiple means through which the students can demonstrate what they have learned [16]. The flexibility and freedom offered by UDL allow the teachers to be innovative in designing inclusive teaching approaches that can suit all students' needs. Moreover, Katz and Porath [17] have identified close a relationship between UDL and Multiple Intelligence (MI) theory, where the latter places great emphasis on recognizing students' strengths, abilities, and challenges, which also adheres to UDL principles. With the assumption that every student possesses a different intellectual composition, Gardner [18] has proclaimed that MI can inform the teachers regarding students' MI profiles. For instance, some students may have well-developed spatial-visual intelligence, while others are more inclined towards verbal-linguistic intelligence. Accordingly, MI forms a strong and relevant theoretical basis for the researchers to develop a UDL-MI-oriented STEM program for rural students.

However, developing an inclusive educational environment using UDL and MI requires teachers to provide appropriate instructional strategies and materials to address every students' learning needs [19]. The application of both the UDL model and MI theory as inclusive learning strategies have proven to be beneficial in meeting the diverse needs of Kenyan primary school students [20]. MI has to a great extent complemented the UDL model by allowing the teachers to understand their students' learning styles prior to designing appropriate teaching approaches, which can later be applied to improve the students' attitudes, and academic achievement [20]. Similarly, the employment of the UDL model and MI-based instruction grants teachers the responsibility to create an alternative learning environment that acknowledges all students regardless of their background and abilities [21].

In the last recent years, UDL has gained wide recognition as a flexible pedagogical model that can provide accessible learning environments for all students to achieve high-quality education [22]. However, this momentous and notable movement barely exerts significant influence on the existing STEM classroom practices. As reported by [10], STEM teaching in many countries, including Malaysia, involves students to learn using

the same material, in the same manner, at the same pace and uses the standardized national instruments for assessing the students' academic achievement. By aligning the STEM teaching practices with UDL principles and MI theory, this innovative approach is able to foster inclusivity by addressing the individual student's uniqueness which ultimately leads to the development of positive attitudes towards STEM [19]. Although UDL has interesting appeal and MI has a conceptually sound foundation to serve as an inclusive pedagogical approach, there is little scientific evidence to support their effectiveness for STEM education. Therefore, this study aims to provide a first step toward empirically testing the UDL-MI-oriented STEM program. The key research questions which guided this study were:

1. Are there significant differences between the UDL-MI-oriented STEM program and traditional STEM program on eighth-grade rural students' attitudes towards STEM?
2. What are the results of post-test and follow-up interviews of the students from both groups with respect to changes in their attitudes towards STEM?

## 2. Materials and Methods

### 2.1. Sampling Approach

The study sample comprised of 122 eighth-grade rural students from a primary school located in East Malaysia. They were selected through the purposive sampling method and randomly distributed into experimental (N = 62) and control groups (N = 60). The experimental group experienced learning through the UDL-MI-oriented STEM program and the control group was taught conventionally in the traditional classrooms. The predetermined criteria included school location within a range of 60–100 km radius of the nearest city center, eighth-grade students of more than 100, and the school science teacher should possess at least a Master's degree in Science Education in addition to a minimum of five-year teaching experience.

### 2.2. Research Design

This study employed a mixed research design of both quantitative and qualitative methods within an embedded experimental model [23]. In this study, the quantitative method aimed to test the effectiveness of the UDL-MI oriented STEM program while the qualitative method acted as complementary evidence to support and elaborate on the quantitative findings. The 10-week quasi-experimental study involved the random assignment of experiment and control groups who were both taught by the same science teacher. The teacher had a Master's degree in Science Education, and six years of teaching experience. Prior to the intervention, the Multiple Intelligences Development Assessment Scales (MIDAS for KIDS: All About Me) [24] was administered to the experimental group to obtain an initial understanding of their MI profiles. MIDAS served as an important standard psychometric procedure to measure individual MI profiles. Proven as one of the most efficient and valid methods for obtaining a rich understanding of one's MI profile, MIDAS provided a powerful foundation for the teacher to design appropriate teaching approaches that could unleash the students' full potential. Advocated by Gardner as meeting the principles of MI assessment, each of the 93 items in MIDAS was constructed to examine how the student applied his or her abilities using real-world activities through the assessment of intellectual disposition, styles, associated skills, and individual preference. These results would help the researchers and teachers in designing a UDL-MI-oriented STEM program based on the students' respective MI profiles. Figure 1 illustrates the research procedure employed for this study.

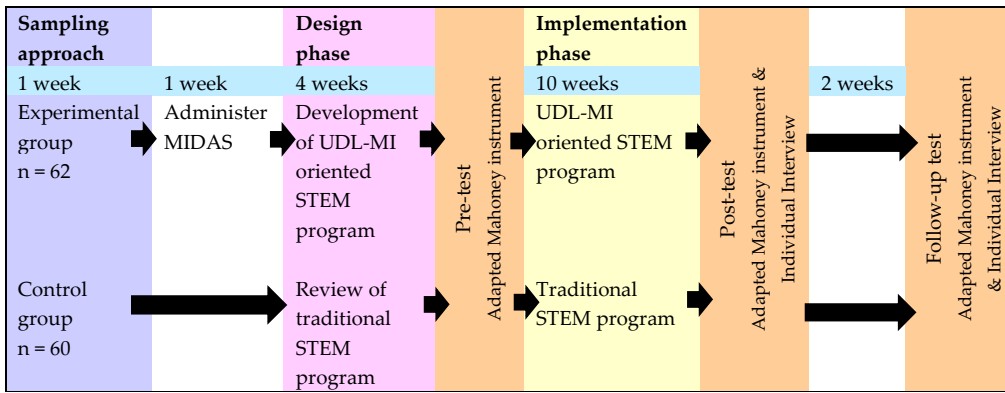

**Figure 1.** Illustration of research procedure. MIDAS = An adapted Multiple Intelligences Development Assessment Scales (MIDAS for KIDS: All About Me), UDL-MI = Universal Design for Learning-Multiple Intelligence, STEM = Science, Technology, Engineering, Mathematics.

*2.3. Development of UDL-MI-Oriented STEM Program*

Informed by the MIDAS results of the experimental group, six experts were asked to review, provide narrative suggestions, and endorse the UDL-MI-oriented STEM program. The experts consisted of two UDL-MI experts, two professors of STEM education, and two senior science teachers. The following were brief descriptions of their respective roles, which were:

1.  identifying specific STEM learning objectives,
2.  addressing UDL core principles through MI lens,
3.  considering the opportunities and possible challenges, and
4.  suggesting appropriate STEM learning units based on UDL principles and MIDAS results.

These experts worked closely to construct STEM learning units on environmental sustainability and conservation. Examples of the learning units were renewable energy, water security, biodiversity protection, agricultural sustainability, and climate change. In designing the UDL-MI-oriented STEM program, the experts highlighted the need to address multiple means of engaging the students throughout the STEM lessons. The experts proposed different teaching approaches that meet the students' MI profiles in order to promote their active involvement in the teaching process. For instance, the use of collaborative group work, experimentation, role-play, and drawing activity. Furthermore, the experts also emphasized multiple representations of knowledge where the use of various learning resources from printed text to visual images was highly encouraged. Finally, the experts gave priority for the teacher to employ multiple and flexible means of expression in providing alternatives for the students to demonstrate what they have learned. Various assessment strategies were also proposed to complement the assessment, which included self-assessment and peer-assessment. The variety of assessment methods used could provide wider perspectives for the students to improve their learning. These careful thoughts offered by the experts remarkably helped the researchers in designing an inclusive UDL-MI-oriented STEM program. Additionally, these experts were responsible for checking the traditional STEM program to ensure that the control group received similar content knowledge, but without the application of UDL and MI principles. Finally, the senior teachers volunteered to carry out classroom observations for both groups in order to ensure that the teaching process was conducted based on the intended STEM programs.

*2.4. Teaching in the Control Group*

Governed by the idea of the teacher as a knowledge provider, the lecture method formed the major part of the instructional activities for the students in the control group. For instance, the teacher began the teaching by explaining the STEM concepts, while the students were listening attentively to the lecture. The remaining time was devoted to the

students to write down the explanations on their exercise books and for the teacher to assess the students using printed worksheets.

### 2.5. Instruments

The Mahoney Student Attitude Toward STEM instrument was adapted to assess the relevant dimensions of students' attitudes [11]. The questionnaire was reduced to 25 items after being examined for content validity by the field experts (n = 2 professors of STEM education, and n = 2 professors of educational psychology). There were four dimensions involved, namely interest in a STEM career (six items), perceived ability in STEM skills (six items), value of STEM (six items), and commitment to STEM learning (seven items). Next, internal consistency analysis and exploratory factor analysis were conducted. Alpha reliability for the whole scale was 0.89, while alpha reliabilities for the sub-scales ranged from 0.87 to 0.96. With reference to the previous research which reviewed 69 science education research with alpha values quotation, it was reported that alpha value of 0.71–0.91 showed good reliability and value of 0.93–0.94 indicated excellent reliability [25]. Therefore, the calculated alpha value for the whole scale, which was 0.89, showed good reliability, while the calculated alpha values for the sub-scales ranging from 0.87 to 0.96 indicated good to excellent reliability. Each response was recorded using a five-point Likert scale, ranging from "1—Strongly disagree", "2—Disagree", "3—Neither agree nor disagree", "4—Agree", and "5—Strongly agree". Table 1 describes the dimensions and items of the adapted instrument.

**Table 1.** Adapted Mahoney Student Attitude Toward Science, Technology, Engineering, and Mathematics (STEM) instrument used in the study.

| Dimension | Item |
|---|---|
| Interest in a STEM career | I would like to have a career in STEM . . . |
| | Scientists make a meaningful difference in the world . . . |
| | Having a career in STEM would be challenging . . . |
| | People will respect me for doing scientific work . . . |
| | Job in STEM-related area is boring . . . |
| | I would enjoy a career in STEM . . . |
| Perceived ability in STEM skills | I am good at STEM projects involving . . . |
| | STEM activities are difficult for me . . . |
| | I perform well in STEM subjects . . . |
| | I am able to understand complex STEM concepts . . . |
| | STEM concept is easy . . . |
| | I struggle to do STEM homework . . . |
| Value of STEM | STEM is important for . . . |
| | STEM has no value to me . . . |
| | Learning STEM will not help me . . . |
| | STEM is worthy to be understood . . . |
| | STEM advancement is good . . . |
| | I believe all should learn STEM . . . |

**Table 1.** *Cont.*

| Dimension | Item |
|---|---|
| **Commitment to STEMlearning** | I would like more advanced STEM subjects in … |
| | I am eager to participate in after-school STEM programs … |
| | I intend to develop my abilities in STEM … |
| | I enjoy the challenge of STEM … |
| | I am interested to discover new ways to apply STEM learning … |
| | I have no interest to learn STEM … |
| | STEM is not important for my future … |

At the end of this study, individual semi-structured interviews were conducted with 40 students from the experimental group, and 40 students from the control group. As this study aimed to discover changes in students' attitudes towards STEM over the course of the study, three main interview questions were developed: (i) What STEM activities were the most enjoyable to you?, (ii) How did your STEM learning experience affect your interest in STEM?, and (iii) In your opinion, did your STEM learning experience help you to consider STEM-related courses in higher studies or pursue a career in STEM-related areas? Next, follow-up interviews with the same group of students using similar interview questions were conducted after two weeks of the post-study interviews. Each individual interview session lasted for about 20 min, was audio-recorded, manually transcribed verbatim, and analyzed.

*2.6. Data Analysis*

Two types of data analysis were performed: (i) Quantitative analysis. Repeated measures of STEM-related attitudinal changes were conducted where each dimension was considered as a dependent variable, while the independent variable was the type of teaching employed (UDL-MI-oriented STEM program and traditional STEM program). The changes in students' attitudes towards STEM (pre-test, post-test, and follow-up) in both groups were determined using a double multivariate repeated measure (MANOVA). All assumptions in terms of normality, linearity, and homogeneity of variance-covariance matrices were met before proceeding with the data analysis. (ii) Qualitative analysis. Post-test (n = 40) and follow-up interviews (n = 40) were conducted among randomly selected students from each group to provide greater insights towards understanding the changes in their attitudes. The follow-up interview had a particular focus to explore the maintenance of attitude change towards STEM among the students. Employing the inductive analytical procedure, this study utilized the five levels of science attitude scale [26] to identify the degree of changes of attitude towards STEM based on five categories, from "none at all" to "a great deal" for the post-test interview. On the other hand, the five categories were revised from "largely negative attitudinal change" to "largely positive attitudinal change" for the follow-up interviews. All qualitative data were coded using the coding system introduced by [27]. These categories were crucial as they allowed the researchers to establish consistency between quantitative and qualitative outcomes based on the embedded experimental model and to minimize possible researcher bias. Within the context of reliability analysis, one of the researchers and one external expert independently analyzed the interview transcripts based on the coding sheets. They later reviewed the interview transcripts together for any discrepancies to reach a consensus on the degree of attitudinal changes following the coding sheets. Percentage agreements of 94% for the post-test interviews and 91% for the follow-up interviews were achieved.

## 3. Results

Repeated measures of attitudes towards STEM—the results of the multivariate analysis of variance (MANOVA) showed statistically significant differences between the control and experimental groups over time for all dependent variables; Wilks' Lambda, $\wedge = 0.41$; $F(10, 112) = 17.98$, $p < 0.0001$, $\eta^2 = 0.64$. The Wilks' Lambda commonly indicated by the symbol "$\wedge$", was a statistic used to examine for any significant differences between the means of groups on a combination of dependent variables [28]. The statistic $\Lambda$ was mathematically transformed to an F-test statistic for obtaining a significance level—the *p*-value. The level of significance of all tests was set at *p*-value < 0.05. The Wilks' Lambda values ranged from 0 to 1, where smaller values indicated larger group dispersion [29]. The multivariate value which was represented by the symbol "$\eta^2$", had a value ranging from 0 to 1, to indicate the multivariate effect size index. The effect size measured for the social science research area was 0.01 = small, 0.06 = medium, and 0.14 = large [30]. So, in this research, there was a high intergroup variability as the value of $\wedge = 0.41$ was closer to 0. The multivariate value which was $\eta^2 = 0.64$ showed a large multivariate effect size of 64%. Table 2 provides the mean scores and standard deviations for each dependent variable pre-test, post-test, and at follow-up. The mean scores of experimental and control groups at the beginning of the study were proximal, indicating that they have similar attitudes towards STEM. However, comparisons for the repeated attitudes variables between the pre-test and the post-test revealed significant increases in favor of the experimental group, where $F(1, 120) = 73.61$, $p < 0.0001$, $\eta^2 = 0.41$ for Interest in a STEM career, $F(1, 120) = 72.89$, $p < 0.0001$, $\eta^2 = 0.39$ for Perceived ability in STEM skills, $F(1, 120) = 69.42$, $p < 0.0001$, $\eta^2 = 0.34$ for Value of STEM, and $F(1, 120) = 70.26$, $p < 0.0001$, $\eta^2 = 0.35$ for Commitment to STEM learning. In terms of the percentage changes, the control and experimental group recorded the value of percentage respectively: 33.5% and 74.6% for Interest in a STEM career, 33.6% and 59.2% for Perceived ability in STEM skills, 27.8% and 66.4% for Value of STEM, and 37.6% and 70.8% for Commitment to STEM learning. Although the data indicated that both groups significantly improved their attitudes towards STEM in all dimensions, an average difference of 34% favored the experimental group.

**Table 2.** Mean scores and standard deviations for repeated measures of science, technology, engineering, and mathematics (STEM) attitudes at pre-test, post-test, and follow-up for both groups.

| Attitudes towards STEM | Control Group (N = 60) | | Experimental Group (N = 62) | |
|---|---|---|---|---|
| | **MS (PD)** | **(SD)** | **MS (PD)** | **(SD)** |
| *Interest in a STEM career* | | | | |
| Pre | 13.98 (33.5%) | (2.35) | 13.56 (74.6%) | (3.41) |
| Post | 18.67 (7.6%) | (3.16) | 23.68 (2.4%) | (2.42) |
| Follow-up | 17.26 | (2.29) | 23.11 | (2.39) |
| *Perceived ability in STEM skills* | | | | |
| Pre | 11.88 (33.6%) | (1.76) | 12.35 (59.2%) | (3.08) |
| Post | 15.87 (7.8%) | (2.82) | 19.66 (1.2%) | (2.01) |
| Follow-up | 14.63 | (2.63) | 19.42 | (2.89) |
| *Value of STEM* | | | | |
| Pre | 12.28 (27.8%) | (1.87) | 11.68 (66.4%) | (2.90) |
| Post | 15.69 (9.4%) | (2.66) | 19.44 (2.2%) | (1.87) |
| Follow-up | 14.21 | (2.76) | 19.01 | (1.79) |
| *Commitment to STEM learning* | | | | |
| Pre | 9.87 (37.6%) | (1.82) | 9.77 (70.8%) | (2.68) |
| Post | 13.58 (4.4%) | (2.13) | 16.69 (2.8%) | (1.74) |
| Follow-up | 12.98 | (2.36) | 16.23 | (1.80) |

1 MS = mean score, PD = percentage decrease, SD = standard deviation.

Comparisons between the post-test and the follow-up showed that the differences between the control and experimental groups were statistically significant for all repeated attitude variables; $F(1, 120) = 10.98$, $p < 0.01$, $\eta^2 = 0.09$ for Interest in a STEM career, $F(1, 120) = 12.86$, $p < 0.01$, $\eta^2 = 0.11$ for Perceived ability in STEM skills, $F(1, 120) = 6.42$, $p < 0.05$, $\eta^2 = 0.04$ for Value of STEM, and $F(1, 120) = 6.11$, $p < 0.05$, $\eta^2 = 0.05$ for Commitment to STEM learning. The tabulated data revealed a decline between the post-test scores and the follow-up scores for both groups. However, the control group significantly lost more attitudes in all attitude dimensions towards STEM when compared to the experimental group. The percentage decreases for the control and experimental groups were respectively as follows: 7.6% and 2.4% for Interest in a STEM career, 7.8% and 1.2% for Perceived ability in STEM skills, 9.4% and 2.2% for Value of STEM, and 4.4% and 2.8% for Commitment to STEM learning. Although the percentage of students' attitudes towards STEM in both groups declined in all attitude dimensions, an average difference of 5.2% between the two groups was calculated, thus favoring the experimental group. Overall, the quantitative results indicated that the UDL-oriented STEM program had significantly improved and maintained positive attitudes of students towards STEM in comparison to the traditional STEM program. Below is the equation used to calculate the percentage decrease.

$$\frac{\text{Mean score (Post} - \text{test)} - \text{Mean score (Follow} - \text{up)} \times 100\%}{\text{Mean score (Post} - \text{test)}} \tag{1}$$

*Qualitative Results*

Figures 2 and 3 illustrate the findings of the post-test and follow-up interview analyses. The analysis of the post-test interviews showed that 20% of the students from the control group had a lot or a great deal of positive changes in their attitude towards STEM. While up to 80% of the students from the experimental group showed similar attitude changes towards STEM (see Figure 2). The analysis of the follow-up interview indicated that 70% of the experimental group students and 10% of the control group students were found to maintain positive attitudes towards STEM. Almost 15% of the experimental group and 70% of the control group were categorized in the types of slightly or largely negative change.

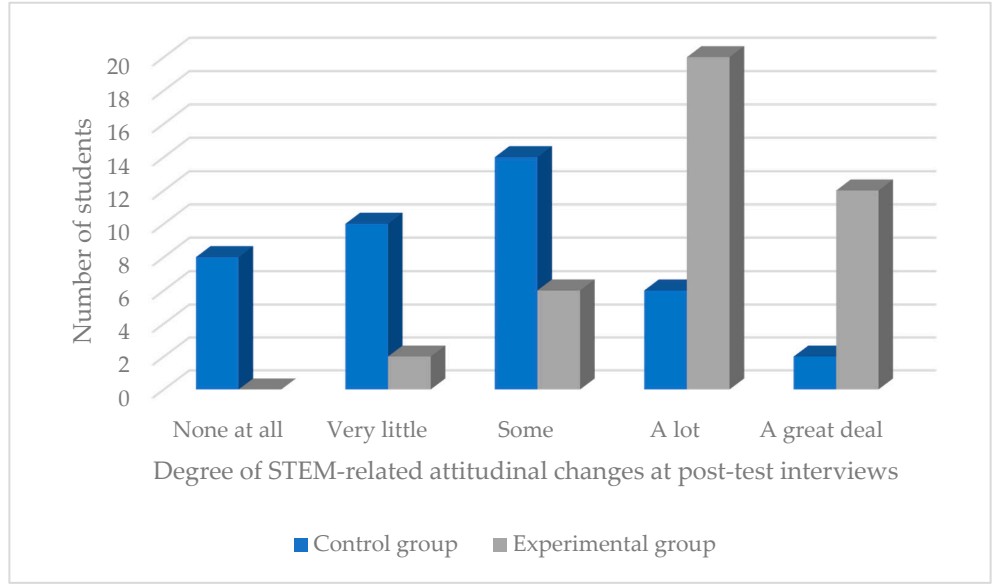

**Figure 2.** The degree of science, technology, engineering, and mathematics (STEM)-related attitudinal changes at post-test interviews for students from control and experimental groups.

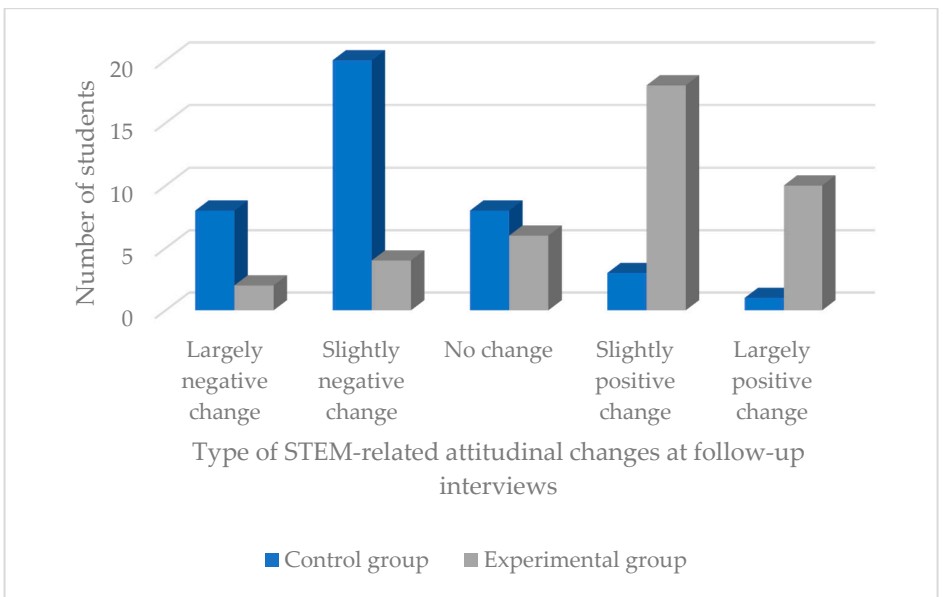

**Figure 3.** The type of science, technology, engineering, and mathematics (STEM)-related attitudinal changes between the time periods of the post-test interview and follow-up interview for students from control and experimental groups.

The qualitative analysis revealed that almost all students in the experimental group were able to distinguish their learning experiences from their previous traditional STEM program and their learning experiences during the UDL-MI-oriented STEM program. Furthermore, a great majority of them reported that the UDL-MI-oriented STEM program remarkably increased their attitudes towards STEM. For instance:

" . . . the new program is very interesting. I do not know that STEM can be very exciting. I am considering to get into the science stream during high school . . . "

" . . . I am surprised that the teacher uses different teaching approaches. Some of them suit my learning strength and weakness as well. STEM is no longer boring. I am more interested to learn about STEM, do well in STEM, so I can become an environment scientist . . . "

In addition, the students from the experimental group reported that they immensely enjoyed learning in the UDL-MI-oriented STEM program. They were surprised by the unusual learning activities which include singing a song, doing role plays, debating with other students, drawing mental images, doing hands-on STEM experiments, working in a small group, and presenting individual STEM projects.

" . . . I never knew that STEM would allow such an enjoyable learning atmosphere. I feel great at the end of the class and I wish that the class would never end . . . "

" . . . The first time when the teacher introduces the role-play activity, I was so surprised. How do you act in STEM? I play the role as a bird and I am flying around the class. I like learning though physical activities. The play is so memorable that I can still remember the facts from the play . . . "

The students from the control group expressed their dissatisfaction with the traditional STEM program. They reported that their learning experiences were non-stimulating and predictable. Their daily learning routines mainly involved listening to the lectures, answering the teacher's questions when asked to do so, reading textbooks, copying the notes, and completing assignments. Furthermore, the repeated use of printed worksheets as an assessment tool created additional stress on the students. These negative feelings had an adverse effect on their attitudes towards STEM.

" ... I seem to be able to guess what and how the teacher will teach the class. There is not much variation in terms of teaching approach. I will start fidgeting after sometime to avoid from falling asleep ... "

" ... do not get me wrong. I am not telling that my teacher is teaching badly. I just think that I feel worse at the end of the class, because I am unable to answer the questions in the worksheets correctly even though I pay full attention during the class ... "

Moreover, the students from the control group believed that the traditional STEM program isolated their participation during the teaching and learning session. The teacher-centered approach created a distance between the teacher and students, thus the students felt that they could not actively engage in the classroom. Therefore, they greatly perceived that their learning process and success greatly depends on their teacher.

" ... the teacher knows what is best for me. So, I just sit and listen ... "

In short, critical analysis of the interview data provided evidence of the synergies between UDL principles and the nature of MI teaching approaches as inclusive pedagogical strategy for STEM learning. The UDL-MI-oriented STEM program which emphasized on 'multiple means of engagement and representation', and 'matching the teaching with the students' abilities and weaknesses', was successful not only in developing inclusive STEM classroom for the rural students but also in sustaining the positive attitudes of students towards STEM.

## 4. Discussion

The current education reform and policy documents [31,32] have either implicitly or explicitly highlighted the need for education to be accessible or inclusive to all students. Based on both the quantitative and qualitative findings of this study, there is strong evidence that an inclusive STEM learning environment can be promoted by implementing the UDL-MI-oriented STEM program when compared to the traditional STEM program. The innovative program has significantly improved and sustained eighth-grade rural students' attitudes towards STEM. The effect size value is in favor of the experimental group, where the value was large for pre-test–post-test comparison and moderate for post-test–follow-up comparison.

Although limited research focuses on improving students' attitudes towards STEM using UDL strategies and MI theory, the findings of this study reaffirm previous findings that students enjoy learning science through various teaching methods [33] and the diversity of the teaching approaches help at least 85% of ninth-grade students to enjoy learning science [33].

Another distinctive feature of this study is the significance of inclusive STEM learning practices for rural students to be meticulously designed and implemented based on the students' strengths and abilities [34]. The empirical evidence from this study could accelerate more research on sustaining students' attitudes towards STEM by incorporating both UDL and MI principles for the development of inclusive STEM education. In fact, it is notable that the UDL-MI-oriented program can assist the teacher to implement enjoyable STEM learning activities, and at the same time sustain the students' positive attitude towards STEM.

In contrast to the traditional STEM program, the experimental group highly acknowledged the interesting teaching approaches that allow them to actively engage with STEM learning and use various ways to represent their ideas based on their individual abilities. Likewise, many UDL and MI scholars [33,35] pointed out that a vast variety of teaching approaches and assessment strategies are crucial to match each students' unique intelligence profile to promote students' active engagement in learning STEM. Finally, the UDL-MI-oriented STEM program has granted freedom to the students by offering wide opportunities for them to express their understanding of STEM. Furthermore, the learner-centered teaching approach especially in the implementation of self- and peer-

assessment provides greater autonomy for the students to take control of their own learning, which profoundly improves their STEM attitude.

## 5. Conclusions

This study has serious implications for practicing inclusive STEM education for rural students. The UDL-MI-oriented STEM program applies the UDL model and the MI theory to help the teachers improve and sustain students' positive attitudes towards STEM. The developed STEM program can serve as an educational catalyst for any teachers to practice UDL principles and MI theory in providing inclusive STEM education to the rural students.

It is important to note and present several limitations in this study. Most of the STEM learning units constructed in this study largely focus on environmental and biological sciences. Thus, the effectiveness of the UDL-MI-oriented program is arguable in terms of its practical application within different STEM disciplines such as engineering and mathematics. Therefore, the researchers suggest future studies to extend and investigate the application of this approach in other STEM disciplines. Next, the findings from this single-institution study are not representative of a larger population. Although the positive findings support the UDL-MI-oriented program as an interesting alternative to the traditional classroom teaching for rural students, it is appropriate to suggest future research to implement this approach in different socio-cultural contexts such as among poor urban, students and immigrant students.

**Author Contributions:** Conceptualization, N.M.N.; methodology, N.M.N. and N.M.R.; formal analysis, N.M.N. and N.M.R.; investigation, N.M.N.; writing—original draft preparation, N.N. and N.M.N.; writing—review and editing, N.N. and M.A.A.T.; visualization, M.A.A.T.; project administration, M.A.A.T. All authors have read and agreed to the published version of the manuscript.

**Funding:** This research was funded by the UNIVERSITI KEBANGSAAN MALAYSIA, grant number GG-2020-013.

**Institutional Review Board Statement:** Ethical review and approval were waived for this study, as this study involves no more than minimal risk to subjects.

**Informed Consent Statement:** Informed consent was obtained from all subjects involved in the study.

**Data Availability Statement:** The data presented in this study are available on request from the corresponding author. The data are not publicly available due to data restriction policy by the grant provider.

**Conflicts of Interest:** The authors declare no conflict of interest.

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
