# Peer review of "A Comparison Study between Universal Design for Learning-Multiple Intelligence (UDL-MI) Oriented STEM Program and Traditional STEM Program for Inclusive Education"

_sustainability, doi:10.3390/su13020554_

Round 1
Reviewer 1 Report
- While I think it is a not suitable that the use of alternatives to the lecture method are desirable in every discipline, based on what I have seen, this message has simply not gotten across to much of the engineering community and perhaps other STEMs disciplines. I don't know whether you want to address this in the introduction or in the conclusions but, of course, it would need to be addressed in a diplomatic manner!
- On Line 114 you indicate that you used purposeful selection and random assignment. Could you be more explicit as to the criteria you used to purposely select participants since readers may be interested in this?
- Line 200--to be consistent with line 206 should indicate "quantitative analysis"
- On line 206, you indicate that interviews were conducted for students who were randomly selected from each group. How many such students were randomly selected from each group?
- On a more general note, I admire your integration of quantitative and qualitative methods. An underlying question is whether you suspected more or less change in pre-post test quantitative measures versus in post-test and follow-up interviews? Did you have reasons to expect differential changes between these two methods of data collection?
Reviewer 2 Report
Dear authors,
Thank you very much for giving me the chance to read your paper "A comparison study between Universal Design for Learning-Multiple Intelligence (UDL-MI) oriented STEM program and traditional STEM program for inclusive education". I think that the topic is interesting.
This article presents a current topic of interest to the scientific community.
Writing and argumentative ability are optimal, which facilitates the understanding and reading of the text.
However, there are a few points to keep in mind: the research methodology is not clear.
It should be explained that the study is based on two researches, one quantitative and one qualitative. A conceptual comparison should also be made between these two types of research.
Finally, the conclusions can be extended by mentioning the limitations of the study.
